# ResAct: Reinforcing Long-term Engagement in Sequential Recommendation with Residual Actor

**Wanqi Xue**[1,*]**, Qingpeng Cai**[2]**, Ruohan Zhan**[3]**, Dong Zheng**[2]**, Peng Jiang**[2]**, Kun Gai**[4]**, Bo An**[1]
[1]Nanyang Technological University, [2]Kuaishou Technology,
[3]Hong Kong University of Science and Technology, [4]Unaffiliated
wanqi001@e.ntu.edu.sg, {caiqingpeng, zhengdong, jiangpeng}@kuaishou.com, rhzhan@ust.hk,
gai.kun@qq.com, boan@ntu.edu.sg

## Abstract

Long-term engagement is preferred over immediate engagement in sequential recommendation as it directly affects product operational metrics such as daily active users (DAUs) and dwell time. Meanwhile, reinforcement learning (RL) is widely regarded as a promising framework for optimizing long-term engagement in sequential recommendation. However, due to expensive online interactions, it is very difficult for RL algorithms to perform state-action value estimation, exploration and feature extraction when optimizing long-term engagement. In this paper, we propose ResAct which seeks a policy that is close to, but better than, the online-serving policy. In this way, we can collect sufficient data near the learned policy so that state-action values can be properly estimated, and there is no need to perform online interaction. ResAct optimizes the policy by first reconstructing the online behaviors and then improving it via a **Res**idual **Act**or. To extract long-term information, ResAct utilizes two information-theoretical regularizers to confirm the expressiveness and conciseness of features. We conduct experiments on a benchmark dataset and a large-scale industrial dataset which consists of tens of millions of recommendation requests. Experimental results show that our method significantly outperforms the state-of-the-art baselines in various long-term engagement optimization tasks.

## 1 Introduction

In recent years, sequential recommendation has achieved remarkable success in various fields such as news recommendation (Wu et al., 2017; Zheng et al., 2018; de Souza Pereira Moreira et al., 2021), digital entertainment (Donkers et al., 2017; Huang et al., 2018; Pereira et al., 2019), E-commerce (Chen et al., 2018; Tang & Wang, 2018) and social media (Zhao et al., 2020b; Rappaz et al., 2021). Real-life products, such as Tiktok and Kuaishou, have influenced the daily lives of billions of people with the support of sequential recommender systems. Different from traditional recommender systems which assume that the number of recommended items is fixed, a sequential recommender system keeps recommending items to a user until the user quits the current service/session (Wang et al., 2019; Hidasi et al., 2016). In sequential recommendation, as depicted in Figure 1, users have the option to browse endless items in one session and can restart a new session after they quit the old one (Zhao et al., 2020c). To this end, an ideal sequential recommender system would be expected to achieve i) low return time between sessions, i.e., high frequency of user visits; and ii) large session length so that more items can be browsed in each session. We denote these two characteristics, i.e., return time and session length, as long-term engagement, in contrast to immediate engagement which is conventionally measured by click-through rates (Hidasi et al., 2016). Long-term engagement is preferred over immediate engagement in sequential recommendation as it directly affects product operational metrics such as daily active users (DAUs) and dwell time.

---

[*]The work was done during an internship at Kuaishou Technology.

Despite great importance, unfortunately, how to effectively improve long-term engagement in sequential recommendation remains largely uninvestigated. Relating the changes in long-term user engagement to a single recommendation is a tough problem (Wang et al., 2022). Existing works on sequential recommendation have typically focused on estimating the probability of immediate engagement with various neural network architectures (Hidasi et al., 2016; Tang & Wang, 2018). However, they neglect to explicitly improve user stickiness such as increasing the frequency of visits or extending the average session length. There have been some recent efforts to optimize long-term engagement in sequential recommendation. However, they are usually based on strong assumptions such as recommendation diversity will increase long-term engagement (Teo et al., 2016; Zou et al., 2019). In fact, the relationship between recommendation diversity and long-term engagement is largely empirical, and how to measure diversity properly is also unclear (Zhao et al., 2020c).

Recently, reinforcement learning has achieved impressive advances in various sequential decision-making tasks, such as games (Silver et al., 2017; Schrittwieser et al., 2020), autonomous driving (Kiran et al., 2021) and robotics (Levine et al., 2016). Reinforcement learning in general focuses on learning policies which maximize cumulative reward from a long-term perspective (Sutton & Barto, 2018). To this end, it offers us a promising framework to optimize long-term engagement in sequential recommendation (Chen et al., 2019). We can formulate the recommender system as an agent, with users as the environment, and assign rewards to the recommender system based on users' response, for example,

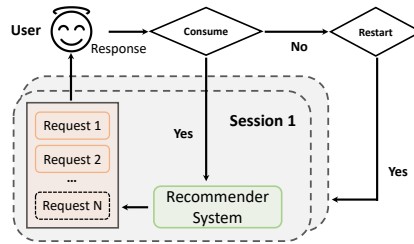

Figure 1: Sequential recommendation.

the return time between two sessions. However, back to reality, there are significant challenges. First, the evolution of user stickiness lasts for a long period, usually days or months, which makes the evaluation of state-action value difficult. Second, probing for rewards in previously unexplored areas, i.e., exploration, requires live experiments and may hurt user experience. Third, rewards of long-term engagement only occur at the beginning or end of a session and are therefore sparse compared to immediate user responses. As a result, representations of states may not contain sufficient information about long-term engagement.

To mitigate the aforementioned challenges, we propose to learn a recommendation policy that is close to, but better than, the online-serving policy. In this way, i) we can collect sufficient data near the learned policy so that state-action values can be properly estimated; and ii) there is no need to perform online interaction. However, directly learning such a policy is quite difficult since we need to perform optimization in the entire policy space. Instead, our method, ResAct, achieves it by first reconstructing the online behaviors of previous recommendation models, and then improving upon the predictions via a **Res**idual **Act**or. The original optimization problem is decomposed into two sub-tasks which are easier to solve. Furthermore, to learn better representations, two information-theoretical regularizers are designed to confirm the expressiveness and conciseness of features. We conduct experiments on a benchmark dataset and a real-world dataset consisting of tens of millions of recommendation requests. The results show that ResAct significantly outperforms previous state-of-the-art methods in various long-term engagement optimization tasks.

## 2 PROBLEM STATEMENT

In sequential recommendation, users interact with the recommender system on a session basis. A session starts when a user opens the App and ends when he/she leaves. As in Figure 1, when a user starts a session, the recommendation agent begins to feed items to the user, one for each recommendation request, until the session ends. For each request, the user can choose to consume the recommended item or quit the current session. A user may start a new session after he/she exits the old one, and can consume an arbitrary number of items within a session. An ideal recommender system with a goal for long-term engagement would be expected to minimize the average return time between sessions while maximizing the average number of items consumed in a session. Formally, we describe the sequential recommendation problem as a Markov Decision Process (MDP) which is defined by a tuple $\langle \mathcal{S}, \mathcal{A}, \mathcal{P}, \mathcal{R}, \gamma \rangle$. $\mathcal{S} = \mathcal{S}_h \times \mathcal{S}_l$ is the continuous state space. $s \in \mathcal{S}$ indicates the state of a user. Considering the session-request structure in sequential recommendation, we decompose $\mathcal{S}$ into two disjoint sub-spaces, i.e., $\mathcal{S}_h$ and $\mathcal{S}_l$, which is used to represent session-level

(high-level) features and request-level (low-level) features, respectively. $\mathcal{A}$ is the continuous action space (Chandak et al., 2019; Zhao et al., 2020a), where $a \in \mathcal{A}$ is a vector representing a recommended item. $\mathcal{P} : \mathcal{S} \times \mathcal{A} \times \mathcal{S} \to \mathbb{R}$ is the transition function, where $p(s_{t+1}|s_t, a_t)$ defines the state transition probability from the current state $s_t$ to the next state $s_{t+1}$ after recommending an item $a_t$. $\mathcal{R} : \mathcal{S} \times \mathcal{A} \to \mathbb{R}$ is the reward function, where $r(s_t, a_t)$ is the immediate reward by recommending $a_t$ at state $s_t$. The reward function should be related to return time and/or session length; $\gamma$ is the discount factor for future rewards.

Given a policy $\pi(a|s) : \mathcal{S} \times \mathcal{A} \to \mathbb{R}$, we define a state-action value function $Q^\pi(s, a)$ which outputs the expected cumulative reward (return) of taking an action $a$ at state $s$ and thereafter following $\pi$:

$$Q^\pi(s_t, a_t) = \mathbb{E}_{(s_{t'}, a_{t'}) \sim \pi} \left[ r(s_t, a_t) + \sum_{t'=t+1}^{\infty} \gamma^{(t'-t)} \cdot r(s_{t'}, a_{t'}) \right]. \tag{1}$$

The optimization objective is to seek a policy $\pi(a|s)$ such that the return obtained by the recommendation agents is maximized, i.e., $\max_\pi \mathcal{J}(\pi) = \mathbb{E}_{s_t \sim d_t^\pi(\cdot), a_t \sim \pi(\cdot|s_t)} [Q^\pi(s_t, a_t)]$. Here $d_t^\pi(\cdot)$ denotes the state visitation frequency at step $t$ under the policy $\pi$.

## 3 REINFORCING LONG-TERM ENGAGEMENT WITH RESIDUAL ACTOR

To improve long-term engagement, we propose to learn a recommendation policy which is broadly consistent to, but better than, the online-serving policy[1]. In this way, i) we have access to sufficient data near the learned policy so that state-action values can be properly estimated because the notorious extrapolation error is minimized (Fujimoto et al., 2019); and ii) the potential of harming the user experience is reduced as we can easily control the divergence between the learned new policy and the deployed policy (the online-serving policy) and there is no need to perform online interaction.

Despite the advantages, directly learning such a policy is rather difficult because we need to perform optimization throughout the entire huge policy space. Instead, we propose to achieve it by first reconstructing the online-serving policy and then improving it. By doing so, the original optimization problem is decomposed into two sub-tasks which are more manageable.

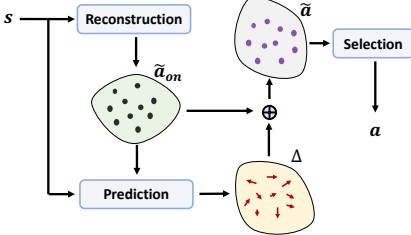

Specifically, let $\hat{\pi}(a|s)$ denote the policy we want to learn; we decompose it into $\hat{a} = a_{on} + \Delta(s, a_{on})$ where $a_{on}$ is sampled from the online-serving policy $\pi_{on}$, i.e., $a_{on} \sim \pi_{on}(a|s)$, and $\Delta(s, a_{on})$ is the residual which is

Figure 2: Workflow of ResAct.

determined by a deterministic actor. We expect that adding the residual will lead to higher expected return, i.e., $\mathcal{J}(\hat{\pi}) \geq \mathcal{J}(\pi_{on})$. As in Figure 2, our algorithm, ResAct, works in three phases:

    i) **Reconstruction:** ResAct first reconstructs the online-serving policy, i.e., $\tilde{\pi}_{on}(a|s) \approx \pi_{on}(a|s)$, by supervised learning. Then ResAct samples $n$ actions from the reconstructed policy, i.e., $\{\tilde{a}_{on}^i \sim \tilde{\pi}_{on}(a|s)\}_{i=1}^n$ as estimators of $a_{on}$;

    ii) **Prediction:** For each estimator $\tilde{a}_{on}^i$, ResAct predicts the residual and applies it to $\tilde{a}_{on}^i$, i.e., $\tilde{a}^i = \tilde{a}_{on}^i + \Delta(s, \tilde{a}_{on}^i)$. We need to learn the residual actor to predict $\Delta(s, \tilde{a}_{on})$ such that $\tilde{a}$ is better than $\tilde{a}_{on}$ in general;

    iii) **Selection:** ResAct selects the best action from the $\{\tilde{a}^i\}_{i=0}^n$ as the final output, i.e., $\arg\max_{\tilde{a}} Q^{\hat{\pi}}(s, \tilde{a})$ for $\tilde{a} \in \{\tilde{a}^i\}_{i=0}^n$.

In sequential recommendation, state representations may not contain sufficient information about long-term engagement. To address this, we design two information-theoretical regularizers to improve the expressiveness and conciseness of the extracted state features. The regularizers are maximizing mutual information between state features and long-term engagement while minimizing the entropy of the state features in order to filter out redundant information. The overview of ResAct is depicted in Figure 3 and we elaborate the details in the subsequent subsections. A formal description for ResAct algorithm is shown in Appendix A.

---

[1]The online-serving policy is a historical policy or a mixture of policies which generate logged data to approximate the MDP in sequential recommendation.

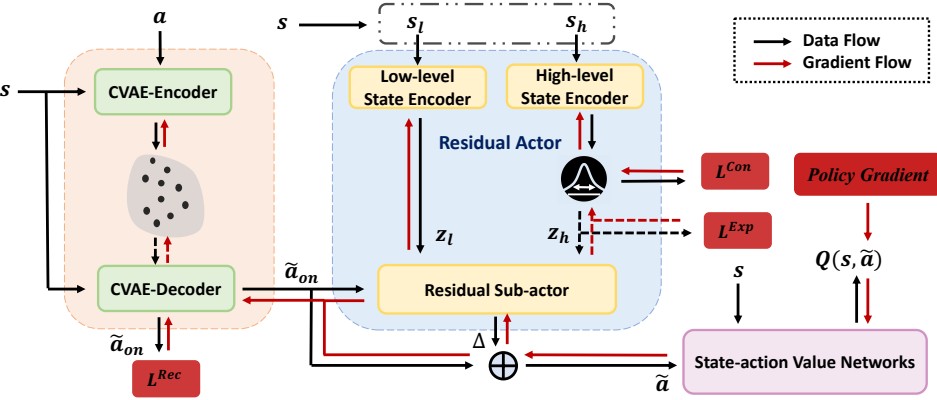

Figure 3: Schematics of our approach. The CVAE-Encoder generates an action embedding distribution, from which a latent vector is sampled for the CVAE-Decoder to reconstruct the action. The reconstructed action $\tilde{a}_{on}$, together with state features extracted by the high-level and low-level state encoders, are fed to the residual actor to predict the residual $\Delta$. After adding the residual, the action and the state are sent to the state-action value networks, from which policy gradient can be generated. The framework can be trained in an end-to-end manner.

## 3.1 RECONSTRUCTING ONLINE BEHAVIORS

To reconstruct behaviors of the online-serving policy, we should learn a mapping $\tilde{\pi}_{on}(a|s)$ from states to action distributions such that $\tilde{\pi}_{on}(a|s) \approx \pi_{on}(a|s)$ where $\pi_{on}(a|s)$ is the online-serving policy. A naive approach is to use a model $D(a|s; \theta_d)$ with parameters $\theta_d$ to approximate $\pi_{on}(a|s)$ and optimize $\theta_d$ by minimizing

$$\mathbb{E}_{s, a_{on} \sim \pi_{on}(a|s)} \left[ (D(a|s; \theta_d) - a_{on})^2 \right]. \tag{2}$$

However, such deterministic action generation only allows for an instance of action and will cause huge deviation if the only estimator is not precise. To mitigate this, we propose to encode $a_{on}$ into a latent distribution conditioned on $s$, and decode samples from the latent space to get estimators of $a_{on}$. By doing so, we can generate multiple action estimators by sampling from the latent distribution. The key idea is inspired by conditional variational auto-encoder (CVAE) (Kingma & Welling, 2014). We define the latent distribution $\mathcal{C}(s, a_{on})$ as a multivariate Gaussian whose parameters, i.e., mean and variance, are determined by an encoder $E(\cdot|s, a_{on}; \theta_e)$ with parameters $\theta_e$. Then for each latent vector $c \sim \mathcal{C}(s, a_{on})$, we can use a decoder $D(a|s, c; \theta_d)$ with parameters $\theta_d$ to map it back to an action. To improve generalization ability, we apply a KL regularizer which controls the deviation between $\mathcal{C}(s, a_{on})$ and its prior which is chosen as the multivariate normal distribution $\mathcal{N}(0, 1)$. Formally, we can optimize $\theta_e$ and $\theta_d$ by minimizing the following loss:

$$L_{\theta_e, \theta_d}^{Rec} = \mathbb{E}_{s, a_{on}, c} \left[ (D(a|s, c; \theta_d) - a_{on})^2 + KL(\mathcal{C}(s, a_{on}; \theta_e) || \mathcal{N}(0, 1)) \right]. \tag{3}$$

where $a_{on} \sim \pi_{on}(a|s)$ and $c \sim \mathcal{C}(s, a_{on}; \theta_e)^2$. When performing behavior reconstruction for an unknown state $s$, we do not know its $a_{on}$ and therefore cannot build $\mathcal{C}(s, a_{on}; \theta_e)$. As a mitigation, we sample $n$ latent vectors from the prior of $\mathcal{C}(s, a_{on})$, i.e., $\{c^i \sim \mathcal{N}(0, 1)\}_{i=0}^n$. Then for each $c^i$, we can generate an estimator of $a_{on}$ by using the decoder $\tilde{a}_{on}^i = D(a|s, c^i; \theta_d)$.

## 3.2 LEARNING TO PREDICT THE OPTIMAL RESIDUAL

By learning the CVAE which consists of $E(\cdot|s, a_{on}; \theta_e)$ and $D(a|s, c; \theta_d)$, we can easily reconstruct the online-serving policy and sample multiple estimators of $a_{on}$ by $\{\tilde{a}_{on}^i = D(a|s, c^i; \theta_d), c^i \sim \mathcal{N}(0, 1)\}_{i=0}^n$. For each $\tilde{a}_{on}^i$, we should predict the residual $\Delta(s, \tilde{a}_{on}^i)$ such that $\tilde{a}^i = \tilde{a}_{on}^i + \Delta(s, \tilde{a}_{on}^i)$ is better than $\tilde{a}_{on}^i$. We use a model $f(\Delta|s, a; \theta_f)$ with parameters $\theta_f$ to approximate the residual function $\Delta(s, a)$. Particularly, the residual actor $f(\Delta|s, a; \theta_f)$ consists of a state encoder and a sub-actor, which are for extracting features from a user state and predicting the residual based

---

[2]$\mathcal{C}(s, a_{on}; \theta_e)$ is parameterized by $\theta_e$ because it is a multivariate Gaussian whose mean and variance are the output of the encoder $E(\cdot|s, a_{on}; \theta_e)$.

on the extracted features, respectively. Considering the bi-level session-request structure in sequential recommendation, we design a hierarchical state encoder consisting of a high-level encoder $f_h(s_h; \theta_h)$ and a low-level encoder $f_l(s_l; \theta_l)$ for extracting features from session-level (high-level) state $s_h$ and request-level (low-level) state $s_l$, respectively. To conclude, the residual actor $f(\Delta|s, a; \theta_f) = \{f_h, f_l, f_a\}$ works as follows:

$$z_h = f_h(s_h; \theta_h), z_l = f_l(s_l; \theta_l); z = Concat(z_h, z_l); \Delta = f_a(z, a; \theta_a). \tag{4}$$

Where $z_h$ and $z_l$ are the extracted high-level and low-level features, respectively; $z$ is the concatenation of $z_h$ and $z_l$, and $f_a(z, a; \theta_a)$ parameterized by $\theta_a$ is the sub-actor. Here, $\theta_f = \{\theta_h, \theta_l, \theta_a\}$.

Given a state $s$ and a sampled latent vector $c \sim \mathcal{N}(0, 1)$, ResAct generates an action with a deterministic policy $\hat{\pi}(a|s, c) = D(\tilde{a}_{on}|s, c; \theta_d) + f(\Delta|s, \tilde{a}_{on}; \theta_f)$. We want to optimize the parameters $\{\theta_d, \theta_f\}$ of $\hat{\pi}(a|s, c)$ so that the expected cumulative reward $\mathcal{J}(\hat{\pi})$ is maximized. Based on the Deterministic Policy Gradient (DPG) theorem (Silver et al., 2014; Lillicrap et al., 2016), we derive the following performance gradients (a detailed derivation can be found in Appendix B):

$$\nabla_{\theta_f} \mathcal{J}(\hat{\pi}) = \mathbb{E}_{s,c} \left[ \nabla_a Q^{\hat{\pi}}(s, a)|_{a=\hat{\pi}(a|s,c)} \nabla_{\theta_f} f(\Delta|s, a; \theta_f)|_{a=D(a|s,c;\theta_d)} \right]. \tag{5}$$

$$\nabla_{\theta_d} \mathcal{J}(\hat{\pi}) = \mathbb{E}_{s,c} \left[ \nabla_a Q^{\hat{\pi}}(s, a)|_{a=\hat{\pi}(a|s,c)} \nabla_{\theta_d} D(a|s, c; \theta_d) \right]. \tag{6}$$

Here $\hat{\pi}(a|s, c) = D(\tilde{a}_{on}|s, c; \theta_d) + f(\Delta|s, \tilde{a}_{on}; \theta_f)$, $p(\cdot)$ is the probability function of a random variable, $Q^{\hat{\pi}}(s, a)$ is the state-action value function for $\hat{\pi}$.

To learn the state-action value function, referred to as *critic*, $Q^{\hat{\pi}}(s, a)$ in Eq. (5) and Eq. (6), we adopt Clipped Double Q-learning (Fujimoto et al., 2018) with two models $Q_1(s, a; \theta_{q_1})$ and $Q_2(s, a; \theta_{q_2})$ to approximate it. For transitions $(s_t, a_t, r_t, s_{t+1})$ from logged data, we optimize $\theta_{q_1}$ and $\theta_{q_2}$ to minimize the following Temporal Difference (TD) loss:

$$L_{\theta_{q_j}}^{TD} = \mathbb{E}_{(s_t, a_t, r_t, s_{t+1})} \left[ (Q_j(s_t, a_t; \theta_{q_j}) - y)^2 \right], j = \{1, 2\};$$
$$y = r_t + \gamma \min \left[ Q_1'(s_{t+1}, \hat{\pi}'(a_{t+1}|s_{t+1}); \theta_{q_1}'), Q_2'(s_{t+1}, \hat{\pi}'(a_{t+1}|s_{t+1}); \theta_{q_2}') \right]. \tag{7}$$

Where $Q_1'$, $Q_2'$, and $\hat{\pi}'$ are target models whose parameters are soft-updated to match the corresponding models (Fujimoto et al., 2018).

According to the DPG theorem, we can update the parameters $\theta_f$ in the direction of $\nabla_{\theta_f} \mathcal{J}(\hat{\pi})$ to gain a value improvement in $\mathcal{J}(\hat{\pi})$:

$$\theta_f \leftarrow \theta_f + \nabla_{\theta_f} \mathcal{J}(\hat{\pi}), \theta_f = \{\theta_h, \theta_l, \theta_a\}. \tag{8}$$

For $\theta_d$, since it also needs to minimize $L_{\theta_e, \theta_d}^{Rec}$, thus the updating direction is

$$\theta_d \leftarrow \theta_d + \nabla_{\theta_d} \mathcal{J}(\hat{\pi}) - \nabla_{\theta_d} L_{\theta_e, \theta_d}^{Rec}. \tag{9}$$

Based on $\hat{\pi}(a|s, c)$, theoretically, we can obtain the policy $\hat{\pi}(a|s)$ by marginalizing out the latent vector $c$: $\hat{\pi}(a|s) = \int p(c) \hat{\pi}(a|s, c) dc$. This integral can be approximated as $\hat{\pi}(a|s) \approx \frac{1}{n} \sum_{i=0}^{n} \hat{\pi}(a|s, c^i)$ where $\{c^i \sim \mathcal{N}(0, 1)\}_{i=0}^{n}$. However, given that we already have a critic $Q_1(s, a; \theta_{q_1})$, we can alternatively use the critic to select the final output:

$$\hat{\pi}(a|s) = \hat{\pi}(a|s, c^*);$$
$$c^* = \arg\max_c Q_1(s, \hat{\pi}(a|s, c); \theta_{q_1}), c \in \{c^i \sim \mathcal{N}(0, 1)\}_{i=0}^{n} \tag{10}$$

### 3.3 FACILITATING FEATURE EXTRACTION WITH INFORMATION-THEORETICAL REGULARIZERS

Good state representations always ease the learning of models (Nielsen, 2015). Considering that session-level states $s_h \in \mathcal{S}_h$ contain rich information about long-term engagement, we design two information-theoretical regularizers to facilitate the feature extraction. Generally, we expect the learned features to have **Expressiveness** and **Conciseness**. To learn features with the desired properties, we propose to encode session-level state $s_h$ into a stochastic embedding space instead of a deterministic vector. Specifically, $s_h$ is encoded into a multivariate Gaussian distribution

$\mathcal{N}(\mu_h, \sigma_h)$ whose parameters $\mu_h$ and $\sigma_h$ are predicted by the high-level encoder $f_h(s_h; \theta_h)$. Formally, $(\mu_h, \sigma_h) = f_h(s_h; \theta_h)$ and $z_h \sim \mathcal{N}(\mu_h, \sigma_h)$ $z_h$ is the representation for session-level state $s_t$. Next, we introduce how to achieve expressiveness and conciseness in $z_h$.

**Expressiveness.** We expect the extracted features to contain as much information as possible about long-term engagement rewards, suggesting an intuitive approach to maximize the mutual information between $z_h$ and $r(s, a)$. However, estimating and maximizing mutual information $I_{\theta_h}(z_h; r) = \iint p_{\theta_h}(z_h) p(r|z_h) \log \frac{p(r|z_h)}{p(r)} dz_h dr$ is practically intractable. Instead, we derive a lower bound for the mutual information objective based on variational inference (Alemi et al., 2017):

$$
\begin{aligned}
I_{\theta_h}(z_h; r) &\geq \iint p_{\theta_h}(z_h) p(r|z_h) \log \frac{o(r|z_h; \theta_o)}{p(r)} dz_h dr; \\
&= \iint p_{\theta_h}(z_h) p(r|z_h) \log o(r|z_h; \theta_o) dz_h dr + H(r),
\end{aligned}
\tag{11}
$$

where $o(r|z_h; \theta_o)$ is a variational neural estimator of $p(r|z_h)$ with parameters $\theta_o$, $H(r) = -\int p(r) \log p(r) dr$ is the entropy of reward distribution. Since $H(r)$ only depends on user responses and stays fixed for the given environment, we can turn to maximize a lower bound of $I_{\theta_h}(z_h; r)$ which leads to the following expressiveness loss (the derivation is in Appendix C):

$$
L_{\theta_h, \theta_o}^{Exp} = \mathbb{E}_{s, z_h \sim p_{\theta_h}(z_h|s_h)} \left[ \mathcal{H}(p(r|s) || o(r|z_h; \theta_o)) \right],
\tag{12}
$$

where $s$ is state, $s_h$ is session-level state, $p_{\theta_h}(z_h|s_h) = \mathcal{N}(\mu_h, \sigma_h)$, and $\mathcal{H}(\cdot||\cdot)$ denotes the cross entropy between two distributions. By minimizing $L_{\theta_h, \theta_o}^{Exp}$, we confirm expressiveness of $z_h$.

**Conciseness.** If maximizing $I_\theta(z_h; r)$ is the only objective, we could always ensure a maximally informative representation by taking the identity encoding of session-level state ($z_h = s_h$) (Alemi et al., 2017); however, such an encoding is not useful. Thus, apart from expressiveness, we want $z_h$ to be concise enough to filter out redundant information from $s_h$. To achieve this goal, we also want to minimize $I_{\theta_h}(z_h; s_h) = \iint p(s_h) p_{\theta_h}(z_h|s_h) \log \frac{p_{\theta_h}(z_h|s_h)}{p_{\theta_h}(z_h)} ds_h dz_h$ such that $z_h$ is the minimal sufficient statistic of $s_h$ for inferring $r$. Computing the marginal distribution of $z_h$, $p_{\theta_h}(z_h)$, is usually intractable. So we introduce $m(z_h)$ as a variational approximation to $p_{\theta_h}(z_h)$, which is conventionally chosen as the multivariate normal distribution $\mathcal{N}(0, 1)$. Since $KL(p_{\theta_h}(z_h)||m(z_h)) \geq 0$, we can easily have the following upper bound:

$$
I_{\theta_h}(z_h; s_h) \leq \iint p(s_h) p_{\theta_h}(z_h|s_h) \log \frac{p_{\theta_h}(z_h|s_h)}{m(z_h)} ds_h dz_h.
\tag{13}
$$

Minimizing this upper bound leads to the following conciseness loss:

$$
\begin{aligned}
L_{\theta_h}^{Con} &= \int p(s_h) \left[ \int p_{\theta_h}(z_h|s_h) \log \frac{p_{\theta_h}(z_h|s_h)}{m(z_h)} dz_h \right] ds_h; \\
&= \mathbb{E}_s \left[ KL(p_{\theta_h}(z_h|s_h) || m(z_h)) \right].
\end{aligned}
\tag{14}
$$

By minimizing $L_{\theta_h}^{Con}$, we achieve conciseness in $z_h$.

## 4 EXPERIMENT

We conduct experiments on a synthetic dataset *MovieLensL-1m* and a real-world dataset *RecL-25m* to demonstrate the effectiveness of ResAct. We are particularly interested in : *Whether ResAct is able to achieve consistent improvements over previous state-of-the-art methods? If yes, why?*

### 4.1 EXPERIMENTAL SETTINGS

**Datasets.** As there is no public dataset explicitly containing signals about long-term engagement, we synthesize a dataset named *MovieLensL-1m* based on *MovieLens-1m* (a popular benchmark for evaluating recommendation algorithms) and collected a large-scale industrial dataset *RecL-25m* from a real-life streaming

Table 1: Statistics of *RecL-25m*.

| | Users | Sessions | Requests |
|---|---|---|---|
| | 99,899 | 6,126,583 | 25,921,753 |
| | **Avg return time (h)** | **Avg session length** | **Avg # of sessions** |
| Mean | - | 4.0449 | 61.3277 |
| 75% | 11.2794 | 4.8792 | 85 |
| 25% | 4.3264 | 2.1358 | 30 |

Table 2: Performance comparison on MovieLensL-1m. The "±" indicates 95% confidence intervals.

| | Return |
|---|---|
| DDPG | 1.7429 ±0.0545 |
| TD3 | 1.7363 ±0.0546 |
| TD3_BC | 1.7135 ±0.0541 |
| BCQ | 1.7898 ±0.0320 |
| IQL | 1.7360 ±0.0546 |
| IL | 1.7485 ±0.0310 |
| IL_CVAE | 1.7344 ±0.0316 |
| ResAct (Ours) | **1.8123 ±0.0319** |

Table 3: Performance comparison on RecL-25m in various tasks. The "±" indicates 95% confidence intervals.

| | Return Time | Session Length | Both |
|---|---|---|---|
| DDPG | 0.6375 ±0.0059 | 0.3290 ±0.0056 | 0.5908 ±0.0092 |
| TD3 | 0.6756 ±0.0133 | 0.4015 ±0.0073 | 0.5498 ±0.0103 |
| TD3_BC | 0.6436 ±0.0059 | 0.3671 ±0.0037 | 0.5563 ±0.0050 |
| BCQ | 0.6837 ±0.0061 | 0.3836 ±0.0033 | 0.5915 ±0.0049 |
| IQL | 0.6296 ±0.0094 | 0.3430 ±0.0057 | 0.5579 ±0.0067 |
| IL | 0.6404 ±0.0058 | 0.3186 ±0.0032 | 0.5345 ±0.0048 |
| IL_CVAE | 0.6410 ±0.0058 | 0.3178 ±0.0031 | 0.5346 ±0.0047 |
| ResAct (Ours) | **0.7980 ±0.0067** | **0.5433 ±0.0045** | **0.6675 ±0.0053** |

platform of short-form videos. *MovieLensL-1m* is constructed by assuming that long-term engagement is proportional to the movie ratings (5-star scale) in *MovieLens-1m*. *RecL-25m* is collected by tracking the behaviors of 99,899 users (randomly selected from the platform) for months and recording their long-term engagement indicators, i.e., return time and session length [3]. The statistics of *RecL-25m* are provided in Table 1, where 25% and 75% denote the corresponding percentile. We did not count the average return time because there are users appearing only once whose return time may go to infinity. The state of a user contains information about gender, age, and historical interactions such as like rate and forward rate. The item to recommend is determined by comparing the inner product of an action and the embedding of videos (Zhao et al., 2020a). Rewards are designed to measure the relative influence of an item on long-term engagement (details are in Appendix D).

**Evaluation Metric and Baselines.** We adopt Normalised Capped Importance Sampling (NCIS) (Swaminathan & Joachims, 2015), a standard offline evaluation method (Gilotte et al., 2018; Farajtabar et al., 2018), to assess the performance of different policies. Given that $\pi_\beta$ is the behavior policy, $\pi$ is the policy to assess, we evaluate the value by

$$\tilde{J}^{NCIS}(\pi) = \frac{1}{|\mathcal{T}|} \sum_{\xi \in \mathcal{T}} \left[ \frac{\sum_{(s,a,r) \in \xi} \tilde{\rho}_{\pi,\pi_\beta}(s,a)r}{\sum_{(s,a,r) \in \xi} \tilde{\rho}_{\pi,\pi_\beta}(s,a)} \right], \quad \tilde{\rho}_{\pi,\pi_\beta}(s,a) = \min \left( c, \frac{\phi_{\pi(s)}(a)}{\phi_{\pi_\beta(s)}(a)} \right). \tag{15}$$

Here $\mathcal{T}$ is the testing set with usage trajectories, $\phi_{\pi(s)}$ denotes a multivariate Gaussian distribution of which mean is given by $\pi(s)$, $c$ is a clipping constant to stabilize the evaluation. We compare our method with various baselines, including classic reinforcement learning methods (DDPG, TD3), reinforcement learning with offline training (TD3_BC, BCQ, IQL), and imitation learning methods (IL, IL_CVAE). Detailed introduction about the baselines are in Appendix E. Our method emphasises on the learning and execution paradigm, and is therefore orthogonal to those approaches which focus on designing neural network architectures, e.g., GRU4Rec (Hidasi et al., 2016).

## 4.2 OVERALL PERFORMANCE

**MovieLensL-1m.** We first evaluate our method on a benchmark dataset *MovieLensL-1m* which contains 1,000,209 anonymous ratings of approximately 3,900 movies made by 6,040 MovieLens users. We sample the data of 5000 users as the training set, and use the data of the remaining users as the test set (with 50 users as the validation set). As in Table 2, our method, ResAct, outperforms all the baselines, indicating its effectiveness. We also provide the learning curve in Figure 4. It can be found that ResAct learns faster and more stable than the baselines on *MovieLensL-1m*.

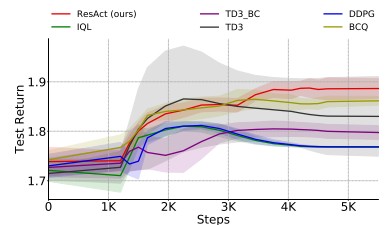

Figure 4: Learning curves of RL-based methods on *MovieLensL-1m*.

**RecL-25m.** We test the performance of ResAct on *RecL-25m* in three modes: i) Return Time mode, where the reward signal $r(\delta)$ is calculated by Eq. 21; ii) Session Length mode, where the reward signal $r(\eta)$ is calculated by Eq. 22; and iii) Both, where reward signal is generated by a convex combination of $r(\delta)$ and $r(\eta)$ with weights of 0.7 and 0.3 respectively. The weights is determined by real-world demands on the operational metrics. We also perform sensitivity analysis on the reward weights in Appendix G. Among the 99,899 users, we randomly selected 80% of the users

---

[3]Data samples and codes can be found in `https://www.dropbox.com/sh/btf0drgm99vmpfe/AADtkmOLZPQ0sTqmsA0f0APna?dl=0`.

as the training set, of which 500 users were reserved for validation. The remaining 20% users constitute the test set. As shown in Table 3, our method significantly outperforms the baselines in all the settings. The classic reinforcement learning algorithms, e.g., DDPG and TD3, perform poorly in the tasks, which indicates that directly predicting an action is difficult. The decomposition of actions effectively facilitates the learning process. Another finding is that the offline reinforcement learning algorithms, e.g., IQL, also perform poorly, even though they are specifically designed to learn from logged data. By comparing with imitation learning, we find that the residual actor has successfully found a policy to improve an action, because behavior reconstruction cannot achieve good performance alone. To compare the learning process, we provide learning curves for those RL-based algorithms. Returns are calculated on the validation set with approximately 30,000 sessions. As in Figure 5, the performance of ResAct increases faster and is more stable than the other methods, suggesting that it is easier and more efficient to predict the residual than to predict an action directly.

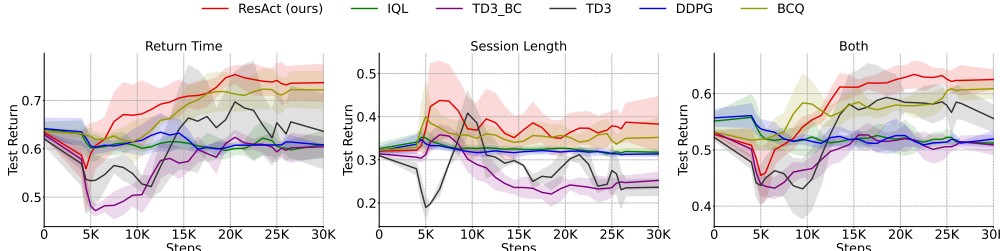

Figure 5: Learning curves of RL-based methods on *RecL-25m*, averaged over 5 runs.

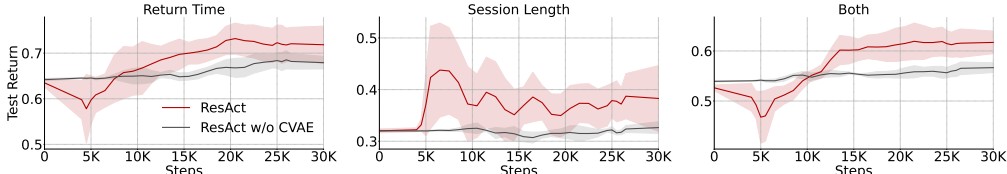

Figure 6: Learning curves for ResAct with CVAE and a deterministic reconstructor (w/o CVAE).

### 4.3 ANALYSES AND ABLATIONS

**How does ResAct work?** To understand the working process of ResAct, we plot t-SNE (Van der Maaten & Hinton, 2008) embedding of actions generated in the execution phase of ResAct. As in Figure 7, the reconstructed actions, denoted by the red dots, are located around the initial action (the red star), suggesting that ResAct successfully samples several online-behavior estimators. The blue dots are the t-SNE embedding of the improved actions, which are generated by imposing residuals on the reconstructed actions. The blue star denotes the executed action of ResAct. We can find that the blue dots are near the initial actions but cover a wider area then the red dots.

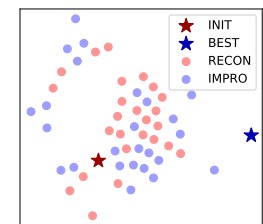

Figure 7: The t-SNE visualization of actions.

**Effect of the CVAE.** We design the CVAE for online-behavior reconstruction because of its ability to generate multiple estimators. To explore the effect of the CVAE and whether a deterministic action reconstructor can achieve similar performance, we disable the CVAE in ResAct and replace it with a feed-forward neural network. The feed-forward neural network is trained by using the loss in Equation 2. Since the feed-forward neural network is deterministic, ResAct does not need to perform the selection phase as there is only one candidate action. We provide the learning curves of ResAct with and without the CVAE in Figure 6. As we can find, there is a significant drop in improvement if we disable the CVAE. We deduce that this is because a deterministic behavior reconstructor can only generate one estimator, and if the prediction is inaccurate, performance will be severely harmed.

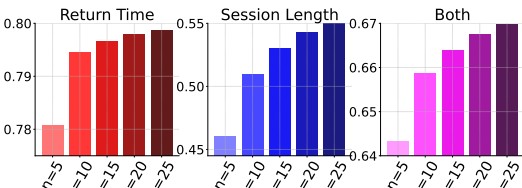

Figure 8: Ablations for the number of online-behavior estimators.

**Number of Online-behavior Estimators.** Knowing that generating only one action estimator might hurt performance, we want to further investigate how the number of estimators will affect the performance of ResAct. We first train a ResAct and then change the number of online-behavior estimators to 5, 10, 15, 20 and 25. As in Figure 8, consistent improvement in performance can be observed across all the three tasks as we increase the number of estimators. The fact suggests that generating more action candidates will benefit the performance, in line with our intuition. We also perform analysis about how the quality of online-behavior estimators could affect the performance in Appendix H. Because the sampling of action estimators is independent, the parallelization of ResAct is not difficult to implement and we can easily speed up the inference.

**Information-theoretical Regularizers.** To explore the effect of the designed regularizers, we disable $L_{\theta_h,\theta_o}^{Exp}$, $L_{\theta_h}^{Con}$ and both of them in ResAct, respectively. As shown in Table 4, the removal of any of the regularizers results in a significant

Table 4: Ablations for the information-theoretical regularizers. The "$\pm$" indicates $95\%$ confidence intervals.

|  | Return Time | Session Length | Both |
|---|---|---|---|
| ResAct | **0.7980** $\pm$**0.0067** | **0.5433** $\pm$**0.0045** | **0.6675** $\pm$**0.0053** |
| w/o $L_{\theta_h,\theta_o}^{Exp}$ | 0.6610 $\pm$0.0060 | 0.3895 $\pm$0.0034 | 0.6074 $\pm$0.0052 |
| w/o $L_{\theta_h}^{Con}$ | 0.6944 $\pm$0.0061 | 0.4542 $\pm$0.0038 | 0.6041 $\pm$0.0051 |
| w/o $L_{\theta_h,\theta_o}^{Exp}$, $L_{\theta_h}^{Con}$ | 0.7368 $\pm$0.0064 | 0.3854 $\pm$0.0033 | 0.6348 $\pm$0.0049 |

drop in performance, suggesting that the regularizers facilitate the extraction of features and thus ease the learning process. An interesting finding is that removing both of the regularizers does not necessarily results in worse performance than removing only one. This suggests that we cannot simply expect either expressiveness or conciseness of features, but rather the combination of both.

## 5 RELATED WORK

**Sequential Recommendation.** Sequential recommendation has been used to model real-world recommendation problems where the browse length is not fixed (Zhao et al., 2020c). Many existing works focused on encoding user previous records with various neural network architectures. For example, GRU4Rec (Hidasi et al., 2016) utilizes Gated Recurrent Unit to exploit users' interaction histories; BERT4Rec (Sun et al., 2019) employs a deep bidirectional self-attention structure to learn sequential patterns. However, these works focus on optimizing immediate engagement like click-through rates. FeedRec (Zou et al., 2019) was proposed to improve long-term engagement in sequential recommendation. However, it is based on strong assumption that recommendation diversity will lead to improvement in user stickiness.

**Reinforcement Learning in Recommender Systems.** Reinforcement learning (RL) has attracted much attention from the recommender system research community for its ability to capture potential future rewards (Zheng et al., 2018; Zhao et al., 2018; Zou et al., 2019; Zhao et al., 2020b; Chen et al., 2021; Cai et al., 2023b;a). Shani et al. (2005) first proposed to treat recommendation as a Markov Decision Process (MDP), and designed a model-based RL method for book recommendation. Dulac-Arnold et al. (2015) brought RL to MDPs with large discrete action spaces and demonstrated the effectiveness on various recommendation tasks with up to one million actions. Chen et al. (2019) scaled a batch RL algorithm, i.e., REINFORCE with off-policy correction to real-world products serving billions of users. Despite the success, previous works required RL agents to learn in the entire policy space. Considering the expensive online interactions and huge state-action spaces, learning the optimal policy in the entire MDP is quite difficult. Our method instead learns a policy near the online-serving policy to achieve local improvement (Kakade & Langford, 2002; Achiam et al., 2017), which is much easier.

## 6 CONCLUSION

In this work, we propose ResAct to reinforce long-term engagement in sequential recommendation. ResAct works by first reconstructing behaviors of the online-serving policy, and then improving the reconstructed policy by imposing an action residual. By doing so, ResAct learns a policy which is close to, but better than, the deployed recommendation model. To facilitate the feature extraction, two information-theoretical regularizers are designed to make state representations both expressive and concise. We conduct extensive experiments on a benchmark dataset *MovieLensL-1m* and a real-world dataset *RecL-25m*. Experimental results demonstrate the superiority of ResAct over previous state-of-the-art algorithms in all the tasks.

**Ethics Statement.** ResAct is designed to increase long-term user engagement, increasing the time and frequency that people use the product. Therefore, it will inevitably cause addiction issues. To mitigate the problem, we may apply some features, e.g., the age of users, to control the strength of personalized recommendation. This may be helpful to avoid addiction to some extent.

**Reproducibility Statement.** We describe the implementation details of ResAct in Appendix F, and also provide our source code and data in the supplementary material and external link.

## ACKNOWLEDGEMENT

This research is supported by the National Research Foundation, Singapore under its Industry Alignment Fund – Pre-positioning (IAF-PP) Funding Initiative. Any opinions, findings and conclusions or recommendations expressed in this material are those of the author(s) and do not reflect the views of National Research Foundation, Singapore.

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

## A    OVERALL ALGORITHM

We provide the learning process of ResAct in Algorithm 1. Particularly, the CVAE is trained to reconstruct the online-serving policy, the residual actor is trained for predicting the optimal residual for each reconstructed action, and the critic networks is trained to guide the optimization of the decoder in CVAE and the residual actor. For the target networks (line 2), $\{\theta_d, \theta_h, \theta_l, \theta_a, \theta_{q_1}\}$ is for the target policy, and $\{\theta_{q_1}, \theta_{q_2}\}$ is for the target critics. The execution process of ResAct is summarized in Algorithm 2. Only the decoder of the CVAE, the residual actor and one of the critics are used during execution.

---

**Algorithm 1:** ResAct-LEARNING

---

**Input:** Logged data collected by the online-serving policy $\mathcal{D} = \{(s_t, a_t, r_t, s_{t+1})\}$

1  Initialize the CVAE: $\{E(c|s, a; \theta_e), D(a|s, c; \theta_d)\}$, the residual actor:
   $\{f_h(z_h|s_h; \theta_h), f_l(z_l|s_l; \theta_l), f_a(\Delta|z, a; \theta_a)\}$, the critic networks
   $\{Q_1(s, a; \theta_{q_1}), Q_2(s, a; \theta_{q_2})\}$, and the variational estimator $o(r|z_h; \theta_o)$

2  Set soft-update rate $\tau$ and initialize the target networks $\theta' \leftarrow \theta$ for $\theta \in \{\theta_d, \theta_h, \theta_l, \theta_a, \theta_{q_1}, \theta_{q_2}\}$

3  **for** $k = 1$ *to* $K$ **do**

4  $\quad$ Sample a batch of transitions $(s_t, a_t, r_t, s_{t+1})$ from $\mathcal{D}$

5  $\quad$ $\theta_e \leftarrow \theta_e - \nabla_{\theta_e} L^{Rec}_{\theta_e, \theta_d}$ ($L^{Rec}_{\theta_e, \theta_d}$ is in Eq. 3)

6  $\quad$ Update $\theta_d$ according to Eq. 9

7  $\quad$ Update $\{\theta_h, \theta_l, \theta_a\}$ according to Eq. 8

8  $\quad$ $\theta_{q_j} \leftarrow \theta_{q_j} - \nabla_{\theta_{q_j}} L^{TD}_{\theta_{q_j}}, j = \{1, 2\}$ ($L^{TD}_{\theta_{q_j}}$ is in Eq. 7)

9  $\quad$ $\theta_h \leftarrow \theta_h - \nabla_{\theta_h} L^{Exp}_{\theta_h, \theta_o} - \nabla_{\theta_h} L^{Con}_{\theta_h}$

10 $\quad$ $\theta_o \leftarrow \theta_o - \nabla_{\theta_o} L^{Exp}_{\theta_h, \theta_o}$
    $\quad$ ($L^{Exp}_{\theta_h, \theta_o}$ is in Eq. 12, and $L^{Con}_{\theta_h}$ is in Eq. 14)

11 $\quad$ Update the target networks:
    $\quad$ $\theta' \leftarrow \tau\theta + (1 - \tau)\theta'$ for $\theta \in \{\theta_d, \theta_h, \theta_l, \theta_a, \theta_{q_1}, \theta_{q_2}\}$

12 **end**

---

---

**Algorithm 2:** ResAct-EXECUTION

---

**Input:** State $s$, number of estimators $n$

**// Reconstruction**

1  Generate $n$ estimators of $a_{on}$: $\{\tilde{a}^i_{on} = D(a|s, c^i; \theta_d), c^i \sim \mathcal{N}(0, 1)\}^n_{i=0}$

**// Prediction**

2  **for** $\tilde{a}_{on} \in \{\tilde{a}^i_{on}\}^n_{i=0}$ **do**

3  $\quad$ Predict the residual $\Delta = f(\Delta|s, \tilde{a}_{on}; \theta_f)$ as in Eq. 4

4  $\quad$ Apply the residual: $\tilde{a} = \tilde{a}_{on} + \Delta$

5  **end**

**// Selection**

6  $a^* = \arg\max_a Q_1(s, a; \theta_{q_1}), a \in \{\tilde{a}^i\}^n_{i=0}$

**Output:** Action $a^*$

---

## B THE DERIVATION OF PERFORMANCE GRADIENTS

We begin by deriving the gradients of $\mathcal{J}(\hat{\pi})$ with respect to the parameters of the residual actor.

$$
\begin{aligned}
\nabla_{\theta_f}\mathcal{J}(\hat{\pi}) &= \iint p(c)p^{\hat{\pi}}(s)\nabla_a Q^{\hat{\pi}}(s,a)|_{a=\hat{\pi}(a|s,c)}\nabla_{\theta_f}\hat{\pi}(a|s,c)\mathrm{d}c\mathrm{d}s \\
&= \iint p(c)p^{\hat{\pi}}(s)\nabla_a Q^{\hat{\pi}}(s,a)|_{a=\hat{\pi}(a|s,c)}\nabla_{\theta_f}f(\Delta|s,a;\theta_f)|_{a=D(a|s,c;\theta_d)}\mathrm{d}c\mathrm{d}s \\
&= \mathbb{E}_{s,c}\left[\nabla_a Q^{\hat{\pi}}(s,a)|_{a=\hat{\pi}(a|s,c)}\nabla_{\theta_f}f(\Delta|s,a;\theta_f)|_{a=D(a|s,c;\theta_d)}\right]
\end{aligned}
\tag{16}
$$

The decoder $D(a|s,c;\theta_d)$ also affects the policy. The gradients of $\mathcal{J}(\hat{\pi})$ with respect to $\theta_d$) is derived similarly:

$$
\begin{aligned}
\nabla_{\theta_d}\mathcal{J}(\hat{\pi}) &= \iint p(c)p^{\hat{\pi}}(s)\nabla_a Q^{\hat{\pi}}(s,a)|_{a=\hat{\pi}(a|s,c)}\nabla_{\theta_d}\hat{\pi}(a|s,c)\mathrm{d}c\mathrm{d}s \\
&= \iint p(c)p^{\hat{\pi}}(s)\nabla_a Q^{\hat{\pi}}(s,a)|_{a=\hat{\pi}(a|s,c)}\nabla_{\theta_d}D(a|s,c;\theta_d)\mathrm{d}c\mathrm{d}s \\
&= \mathbb{E}_{s,c}\left[\nabla_a Q^{\hat{\pi}}(s,a)|_{a=\hat{\pi}(a|s,c)}\nabla_{\theta_d}D(a|s,c;\theta_d)\right]
\end{aligned}
\tag{17}
$$

## C DERIVING THE EXPRESSIVENESS LOSS

We expect the extracted features to contain as much information as possible about long-term engagement rewards, suggesting an intuitive approach to maximize the mutual information between $z_h$ and $r(s,a)$. The mutual information $I_{\theta_h}(z_h; r)$ is defined according to

$$
\begin{aligned}
I_{\theta_h}(z_h; r) &= \iint p(z_h, r)\log\frac{p(z_h, r)}{p(z_h)p(r)}\mathrm{d}z_h\mathrm{d}r \\
&= \iint p_{\theta_h}(z_h)p(r|z_h)\log\frac{p(r|z_h)}{p(r)}\mathrm{d}z_h\mathrm{d}r
\end{aligned}
\tag{18}
$$

However, estimating and maximizing mutual information is practically intractable. Inspired by variational inference (Alemi et al., 2017), we derive a tractable lower bound for the mutual information objective. Considering that $KL(p(r|z_h)||q(r|z_h)) \geq 0$, by the definition of KL-divergence, we have $\int p(r|z_h)\log p(r|z_h)\mathrm{d}r \geq \int p(r|z_h)\log q(r|z_h)\mathrm{d}r$ where $q(r|z_h)$ is an arbitrary distribution. Here, we introduce $o(r|z_h; \theta_o)$ as a variational neural estimator with parameters $\theta_o$ of $p(r|z_h)$. Then,

$$
\begin{aligned}
I_{\theta_h}(z_h; r) &\geq \iint p_{\theta_h}(z_h)p(r|z_h)\log\frac{o(r|z_h; \theta_o)}{p(r)}\mathrm{d}z_h\mathrm{d}r \\
&= \iint p_{\theta_h}(z_h)p(r|z_h)\log o(r|z_h; \theta_o)\mathrm{d}z_h\mathrm{d}r + H(r)
\end{aligned}
\tag{19}
$$

where $H(r) = -\int p(r)\log p(r)\mathrm{d}r$ is the entropy of reward distribution. Since $H(r)$ only depends on user responses and stays fixed for the given environment, we can turn to maximize a lower bound of $I_{\theta_h}(z_h; r)$ which leads to the following expressiveness loss:

$$
\begin{aligned}
L_{\theta_h, \theta_o}^{Exp} &= -\iint p_{\theta_h}(z_h)p(r|z_h)\log o(r|z_h; \theta_o)\mathrm{d}z_h\mathrm{d}r \\
&= -\iiint p(s)p_{\theta_h}(z_h|s)p(r|s, z_h)\log o(r|z_h; \theta_o)\mathrm{d}s\mathrm{d}z_h\mathrm{d}r \\
&= -\iiint p(s)p_{\theta_h}(z_h|s_h)p(r|s)\log o(r|z_h; \theta_o)\mathrm{d}s\mathrm{d}z_h\mathrm{d}r \\
&= \mathbb{E}_{s,z_h\sim p_{\theta_h}(z_h|s_h)}\left[-\int p(r|s)\log o(r|z_h; \theta_o)\mathrm{d}r\right] \\
&= \mathbb{E}_{s,z_h\sim p_{\theta_h}(z_h|s_h)}\left[\mathcal{H}(p(r|s)||o(r|z_h; \theta_o))\right]
\end{aligned}
\tag{20}
$$

where $s$ is state, $s_h$ is session-level state, $p_{\theta_h}(z_h|s_h) = \mathcal{N}(\mu_h, \sigma_h)$, and $\mathcal{H}(\cdot||\cdot)$ denotes the cross entropy between two distributions. By minimizing $L_{\theta_h, \theta_o}^{Exp}$, we confirm expressiveness of $z_h$.

# D DATASETS

## D.1 *MovieLensL-1m*

*MovieLensL-1m* is synthesized from *MovieLens-1m* which is representative benchmark dataset for sequential recommendation. *MovieLens-1m* provides 1,000,209 anonymous ratings of approximately 3,900 movies made by 6,040 MovieLens users. Ratings are made on a 5-star scale. As *MovieLens-1m* does not contain any information about long-term user engagement, to generalize it to long-term engagement problem, we make an assumption that a user's long-term engagement is proportional to the movie ratings. Specifically, we assume that recommending a movie for which a user rates 2-stars will not affect engagement, a movie with 3-stars, 4-stars and 5-stars will benefit the long-term engagement by 1, 2, and 3, respectively. Recommending a movie with 1-star is harmful to engagement and will be given a negative reward, -1. The task in *MovieLensL-1m* is to maximize cumulative benefits on long-term engagement.

## D.2 DESIGNING OF REWARDS IN *RecL-25m*

The rewards of long-term engagement in *RecL-25m* are designed based on the statistics of the dataset. As a general guideline, we expect rewards to reflect the influence of recommending an item on a user. However, behaviors of users have large variance which makes the influence difficult to measure. For example, if we simply make rewards proportional to session length, or inversely proportional to return time, the recommender system would focus on improving the experience of high activity users, because by doing so it can obtain larger rewards. However, in reality, it is equally if not more important to facilitate the conversion of low activity user to high activity user, which requires us to improve the experience of low activity users. To address this issue, we turn to measuring the **relative** influence of an item. Concretely, we calculate the average return time $\delta_{avg}^u$ and the average session length $\eta_{avg}^u$ for a user $u$, and use these two statistics to quantify rewards. For user $u$, given a time duration $\delta^u$ between two sessions, the corresponding reward is calculated by

$$r(\delta^u) = \left( \lfloor \frac{\min(\delta_{avg}^u, \delta_{75\%})}{\delta^u} \rfloor \right).clip(0,5) \tag{21}$$

where $\delta_{75\%}$ is the 75th percentile of the average return time for all users, which is designed to differentiate active users and inactive users. Rewards for the session length is calculated similarly as

$$r(\eta^u) = \left( \lfloor \frac{\eta^u}{\eta_{avg}^u \times 0.8} \rfloor \right).clip(0,5) \tag{22}$$

where $\eta^u$ is the length of a session in the logged data of user $u$. Since $\delta^u$ and $\delta$ can only be calculated at session-level, without loss of generality, we provide rewards at the end of each session, where rewards for return time is assigned to the previous session.

# E BASELINES

Our method is compared with various baselines, including classic reinforcement learning methods (DDPG, TD3), offline reinforcement learning algorithms (TD3_BC, BCQ, IQL), and imitation learning methods (IL, IL_CVAE):

- **DDPG** (Lillicrap et al., 2016): An reinforcement learning algorithm which concurrently learns a Q-function and a policy. It uses the Q-function to guide the optimization of the policy.
- **TD3** (Fujimoto et al., 2018): An off-policy reinforcement learning algorithm which applies clipped double-Q learning, delayed policy updates, and target policy smoothing.
- **TD3_BC** (Fujimoto & Gu, 2021): An reinforcement learning designed for offline training. It adds a behavior cloning (BC) term to the policy update of TD3.
- **BCQ** (Fujimoto et al., 2019): An off-policy algorithm which restricts the action space in order to force the agent towards behaving similar to on-policy.
- **IQL** (Kostrikov et al., 2022): An offline reinforcement learning method which takes a state conditional upper expectile to estimate the value of the best actions in a state.

- **IL**: Imitation learning treats the training set as expert knowledge and learns a mapping between observations and actions under demonstrations of the expert.
- **IL_CVAE** (Kingma & Welling, 2014): Imitation learning method with the policy controlled by a conditional variational auto-encoder.

## F  EXPERIMENTAL DETAILS

Across all methods and experiments, for fair comparison, each network generally uses the same architecture (3-layers MLP with 256 neurons at each hidden layer) and hyper-parameters. We provide the hyper-parameters for ResAct in Table 5. All methods are implemented with PyTorch.

Table 5: Hyper-parameters of ResAct.

| Hyper-parameter | Value |
|---|---|
| Optimizer | Adam (Kingma & Ba, 2014) |
| Actor Learning Rate | $5 \times 10^{-6}$ |
| Critic Learning Rate | $5 \times 10^{-5}$ |
| Batch Size | 4096 |
| Normalized Observations | Ture |
| Gradient Clipping | False |
| Discount Factor | 0.9 |
| Number of Behavior Estimators | 20 |
| Weight of $L^{Exp}$ | $5 \times 10^{-2}$ |
| Weight of $L^{Con}$ | $5 \times 10^{-1}$ |
| Target Update Rate | $1 \times 10^{-2}$ |
| Number of Epoch | 5 |

## G  SENSITIVITY ANALYSIS FOR THE REWARD WEIGHTS

When setting the reward weights in the Both mode, we use some usual empirical values by following the real-world requirements for operational metrics. The reward occurs only at the end of each session, which makes it representative for sequential recommendations. If we try other designs, only the value of the reward will change, not the frequency of learning signal. To justify that our algorithm is robust to different reward weights, we perform sensitivity analysis for the weights (return time: session length) of rewards in the Both mode. As shown in Table 6, our algorithm consistently outperforms the baselines under different reward weights.

Table 6: Sensitivity Analysis for the Reward Weights in the Both Mode.

| | (0.7: 0.3) | (0.5: 0.5) | (0.3: 0.7) |
|---|---|---|---|
| DDPG | 0.5908 ±0.0092 | 0.5040 ±0.0073 | 0.4172 ±0.0059 |
| TD3 | 0.5498 ±0.0133 | 0.4941 ±0.0086 | 0.4385 ±0.0076 |
| TD3_BC | 0.5563 ±0.0050 | 0.4978 ±0.0043 | 0.4393 ±0.0038 |
| BCQ | 0.5915 ±0.0049 | 0.5261 ±0.0042 | 0.4605 ±0.0037 |
| IQL | 0.5579 ±0.0067 | 0.4812 ±0.0054 | 0.4046 ±0.0044 |
| IL | 0.5345 ±0.0048 | 0.4727 ±0.0041 | 0.4111 ±0.0036 |
| IL_CVAE | 0.5346 ±0.0047 | 0.4726 ±0.0041 | 0.4107 ±0.0036 |
| ResAct (Ours) | **0.6675 ±0.0053** | **0.5948 ±0.0045** | **0.5220 ±0.0039** |

## H   QUALITY OF ONLINE-BEHAVIOR ESTIMATORS

Despite the selection phase, the quality of the online-behavior estimators, or the action candidates, still significantly affects the performance. On the one hand, the action candidate directly constitutes the final action. On the other hand, the sampled action candidates serve as inputs of the residual module and the selection module. It is certain that sampling an infinite number of action candidates will cover the best action which is sampled by the CVAE. However, action candidates which are far from online-behavior policy may be incorrectly selected. The reason is that the residual module and the selection module are unlikely to encounter such out-of-distribution (OOD) actions and therefore cannot make accurate predictions. The distribution of action candidates should be as close as possible to the distribution of online services to ensure that the output of the residual and selection modules is reliable. We conduct experiments by uniformly sampling 20 action candidates and adding them to the action candidates reconstructed by the CVAE. As in Table 7, there is a significant decrease in performance even though we increase the number of action candidates.

Table 7: Performance comparison between ResAct and ResAct with uniformly augmented action candidates. The "$\pm$" indicates $95\%$ confidence intervals.

|  | Return Time | Session length | Both |
| --- | --- | --- | --- |
| ResAct | 0.7980 $\pm$0.0067 | 0.5433 $\pm$0.0045 | 0.6675 $\pm$0.0053 |
| ResAct + 20 candidates (uniform) | 0.5501 $\pm$0.0068 | 0.3489 $\pm$0.0041 | 0.4839 $\pm$0.0054 |

