# OpenReview forum: "ResAct: Reinforcing Long-term Engagement in Sequential Recommendation with Residual Actor"
_ICLR.cc/2023/Conference — ICLR 2023 poster_

### Official Review · Reviewer_VU33 · 2022-10-23

**Confidence:** 4
**Correctness:** 4
**Technical Novelty And Significance:** 3
**Empirical Novelty And Significance:** 3
**Recommendation:** 8

**Clarity, Quality, Novelty And Reproducibility:**

The paper is well-written and easy to follow. Complete codes and data samples are provided.


**Strength And Weaknesses:**

Strengths:
1.	The motivation of the method is sound. Improving long-term engagement is a difficult problem and directly optimize it is challenging. ResAct decomposes the problem into several manageable sub-tasks which significantly eases the learning process. The idea of first reconstructing and then improving could inspire people to better solve the problem.
2.	The techniques used in this paper is interesting, e.g., the two information-theoretical regularizers. I am impressed that improving expressiveness or conciseness alone will not lead to improvements and only optimizing them together will improve performance.
3.	Thorough experiments are conducted on various of datasets, including a public benchmark dataset and a real-world industrial dataset. Results are compared with a wide range of baselines. The empirical analysis investigates the working process of the method.

Weaknesses:
1.	Some figures can be better clarified, e.g., it is not clear what the dashed line stands for in fig.3
2.	As discussed in section 4.3, the inference speed is one of the main limitations of the method.

**Summary Of The Paper:**

The paper introduces a new framework ResAct for improving long-term user engagement in sequential recommendation. ResAct works in three phases: i) Reconstruction - Training a model to imitate the current recommendation policy and reconstruct recommendation actions; ii) Prediction - Predicting actor residuals that is likely to improve the reconstructed actions in phase one; iii) Selection – selecting the optimal action among all improved actions in phase two. The overall framework is interesting and reasonable. I think it can bring new thinking to the community about how to improve long-term engagement in recommendation and might lead to real-world applications.

**Summary Of The Review:**

Overall, the paper has good motivation and the method is novel and interesting. Experiments demonstrate the effectiveness of the method. I support to accept the paper.

---

> ### Author Response · Authors · 2022-11-13
> **To Reviewer VU33**
>
> Thank you for your helpful and insightful comments. Please kindly find our responses below.
>
> ***Q1: Some figures can be better clarified, e.g., it is not clear what the dashed line stands for in Fig.3.***
>
> The dashed lines stand for sampling operations which are used in the reparameterization trick. Thank you for pointing this out. We will explain it in more detail.

---

### Official Review · Reviewer_2DQo · 2022-10-28

**Confidence:** 3
**Correctness:** 3
**Technical Novelty And Significance:** 3
**Empirical Novelty And Significance:** 3
**Recommendation:** 8

**Clarity, Quality, Novelty And Reproducibility:**

Clarity- Paper is well written and understandable by someone familiar with popular RL techniques and recommender systems.
Quality - Experiments appear to be performed and evaluated competently.
Novelty- Authors propose multiple innovations and demonstrate the value of each.
Reproducibility - Work should be highly reproducible, though code and data will only be open-sourced pending acceptance.

**Strength And Weaknesses:**

Strengths -
1. Detailed offline experiments with thorough ablation studies, showing how and why their approach improves over existing work.
2. How to wield RL for recommender systems is an important yet unsolved problem with large real-world impact.
3. Experiments are open-source and open-data (pending paper acceptance).

Weaknesses -
1. The ResAct algorithm and the feature selection regularization are a bit orthogonal to each other, and could potentially be presented in separate publications.
2. It's preferable to use a temporal holdout rather than a population holdout for evaluating a recommender system. In the real world, we train in the past and deploy on the future.
3. The experiments on the MovieLens dataset use simulated data inspired by the real-world open source dataset, so may not reflect actual user behavior.

**Summary Of The Paper:**

This paper investigates how to use reinforcement learning (RL) for sequential recommendations. The authors present several innovations for how to address challenges in recommender systems. For one, exploration with actual users can be expensive as they can lead to bad user experience. Also, due to sparse feedback at long time intervals, it is challenging to perform feature selection to remove noisy inputs. To address the first challenge, the authors developed ResAct, which uses supervised learning to learn the production policy and sample additional actions from that policy to select the best one. This eliminates the need to directly explore with users. To address feature selection, they use information theoretic principles to select features that highly predict the reward without retaining redundant information about the state. They demonstrate gains from their approach against several strong baselines using open-source datasets. They also perform ablation experiments to show what design choices lead to their gains.

**Summary Of The Review:**

I am inclined to accept this paper for publication. The problem of how to apply reinforcement learning to sequential recommendations is an important and unsolved real-world problem. The authors make good progress on this problem and present multiple innovations that improve performance on this problem. They present ablation studies which identify how their proposed algorithms lead to improvements in their offline experiments. Finally, the code and data samples are promised to be open-sourced after publication. I have only a mild concern that the two areas for innovation (ResAct and information theoretic regularization) seem to be orthogonal improvements, but I still think the community will benefit from reading this manuscript.

---

> ### Author Response · Authors · 2022-11-13
> **To Reviewer 2DQo**
>
> Thank you for your insightful and detailed comments. Please kindly find our responses below.
>
> ***Q1:The ResAct algorithm and the feature selection regularization are a bit orthogonal to each other.***
>
> The ResAct together with the information-theoretical regularizers are designed based on the session-request structure in sequential recommendation. We propose to relate state features to long-term engagement learning signals, which will ease the learning process of the modules in ResAct.
>
> ***
>
> ***Q2: It's preferable to use a temporal holdout rather than a population holdout for evaluating a recommender system.***
>
> We agree that a temporal holdout is a realistic setting. We adopt a population holdout because the evolution of long-term engagement could last for a long period and the evolving pattern is relatively stable over time.

---

### Official Review · Reviewer_EPzq · 2022-11-03

**Confidence:** 4
**Correctness:** 4
**Technical Novelty And Significance:** 3
**Empirical Novelty And Significance:** 3
**Recommendation:** 8

**Clarity, Quality, Novelty And Reproducibility:**

Clarity. The paper is very clear and well written. All notions (from RL and Bayesian inference) are well explained.

Quality. The paper is scientifically solid, and the experiments are convincing.

Novelty. This work is original. Integrating residual actor seems to be a good way to learn the policy.

Reproducibility. Experiments seem to be reproducible.


**Strength And Weaknesses:**

The paper is very pleasant to read even if it combines ideas from different fields (reinforcement learning and Bayesian inference). All the different parts are presented concisely and well illustrated, making the paper accessible for the whole community. Integrating residual improvement in this framework seems like a good idea, which makes a good trade off between exploitation and exploration in recommender systems. The paper is scientifically solid, and the experiments are convincing.

**Summary Of The Paper:**

The authors are interested in the long-term engagement of users in recommender systems, which is less studied than immediate engagement. To optimize this long-term engagement in sequential recommendation, they resort reinforcement learning (RL). However, directly applying RL methods to recommender systems can be expensive, and may hurt user experience. To alleviate these issues, the authors introduce ResAct. ResAct is composed of two main modules: (i) the first which reconstructs the online-serving policy, (ii) the second which residually modify the recommendation in order to improve long-term engagement. By doing this, ResAct is able to learn a policy which improves long-term engagement, but stays close to the online-serving policy (limiting the risk of harming user experience).
Moreover, the authors introduce two information-theoretical regularizers (expressiveness and conciseness) which allows to extract meaningful features from sparse reward signals.

**Summary Of The Review:**

This paper is very pleasant to read. The content of the paper is original and scientifically solid.

---

> ### Author Response · Authors · 2022-11-13
> **To Reviewer EPzq**
>
> Thank you for your careful reading and detailed evaluation of our paper. We sincerely appreciate your positive comments!

---

### Official Review · Reviewer_pqv6 · 2022-11-04

**Confidence:** 4
**Correctness:** 2
**Technical Novelty And Significance:** 2
**Empirical Novelty And Significance:** 2
**Recommendation:** 5

**Clarity, Quality, Novelty And Reproducibility:**

The paper is well-organized and logic is clear. But for the details of the proposed solution, more explanation would be appreciated to make the paper more clear. For the proposed solution, the novelty is somehow limited. Without the motivation/intuition of the adapted techniques, the framework looks like a combination of several methods.

**Strength And Weaknesses:**

This paper focuses on an interesting problem in reinforcement learning, where the online exploration is expensive to perform. And the paper is well-organized, the logic and the descriptions of the technique details are clear and easy to follow. The proposed model is kind of interesting, e.g., adding the information related regularizations. The authors also performed comprehensive experiments to support the proposed solution.

However, my concern of the paper lies in the following aspects.
First, the novelty of the proposed solution is somehow limited. The proposed solution performs an offline policy learning, and combines the residual learning and regularization. The method is somehow straightforward and there's not much technique parts really related to the long-term action modeling.

Second, the authors claims that no exploration was needed for the policy. This is based on their assumption of the access to the sufficient data (correct me if I am wrong). This assumption is not valid especially for real-world scenarios. Meanwhile, for reinforcement learning with online exploration, the insufficient data is the key problem to solve. If sufficient data is accessible, why we need a complicated model to do the predication?

Besides, I am not very clear about the motivation/intuition of adoption a residual actor in the framework. Why and how the reward is decomposed into two parts. If we need the residual modeling, does it mean the first reconstruction step is inaccurate, which seems contradict to the assumption that sufficient data is accessible.

For the experiment part, the authors also compared the model with classical reinforcement learning solutions, are they performing any exploration in the experiment? If not, is the comparison valid?

**Summary Of The Paper:**

This paper works on the sequential recommendation for long-term engagement. This topic is interesting and very important in either academia or industrial.
In this work, the authors argued that though reinforcement learning was widely accepted as a standard method for long-term engagement optimization, it is not easy to apply RL in practice due to the requirement of online interaction/serving and model update (feature extraction). The authors proposed a new policy that learns the state-action values without the requirement of exploration. The framework first reconstructed the online behaviors based on the offline data, and then improves the model with residual actor. Further, the framework is enhanced with two information theoretical regularizers which improves the expressiveness and conciseness of the extracted features. Comprehensive experiments are conducted on one two datasets, and the performance is improved compared to the existing baselines.

**Summary Of The Review:**

Overall, the paper works on an interesting problem and proposed an empirically good framework for solving the long-term engagement recommendation problem. However, the details of the framework, especially the motivation and intuition of the adapted techniques, are lacking. For the current version of the paper, the proposed framework is like a combination of existing technique with strong assumption of the data. Therefore, I would suggest a borderline reject for this paper.

---

> ### Author Response · Authors · 2022-11-13
> **To Reviewer pqv6**
>
> Thank you for the detailed and helpful evaluation of our paper. Please kindly find our responses below.
>
> ***Q1: The novelty of the proposed solution is somehow limited. Without the motivation and intuition of the adapted techniques, the framework looks like a combination of several methods.***
>
> Unlike existing RL-based recommender systems which attempt to perform policy optimization in the huge policy space with limited exploration, we are the first to propose to generate the recommendation policy by improving upon a reconstructed policy, which significantly improves the realizability and eases the learning. Our insight is that directly learning a good policy for long-term engagement from interaction data is difficult, but we can turn to first reconstruct a policy from the interaction data and then improve a reconstructed policy through reinforcement learning. The paradigm is distinct from existing methods, and the techniques used in ResAct are designed to address the challenges for realizing this idea. First, because training a policy from scratch by RL is difficult and unstable, we propose to leverage historical interaction data to mimic the behaviors of the online-serving policy, generating a reconstructed policy. Second, improving the recommendation policy in the huge policy space will suffer from the extrapolation error. Therefore, we propose to refine the reconstructed policy in its nearby region with the residual module, by predicting the most promising adjustments for recommendation actions. Third, the learning signals related to long-term engagement are delayed and sparse, then we design the information-theoretical regularizers to relate state features to long-term engagement learning signals, which in turn eases the learning process.
>
> ***
>
> ***Q2: The authors claim that no exploration was needed for the policy, which is based on their assumption of the access to the sufficient data.  If sufficient data is accessible, why do we need a complicated model to do the predication?***
>
> No exploration needed is because we optimize the policy around the region where the online-serving policy is located, rather than optimizing in the whole policy space. The safe historical interaction data provide the diversity we need to refine the recommendation policy.
>
> With high-quality interaction data, which can be collected at scale, we are able to obtain sufficient knowledge about how users will react to policies near the online-service policy. However, directly learning the policy with such knowledge is still difficult (see the performance of DDPG or TD3); simply imitating the recommendation policy from the interaction data can at most learn a policy which is close to the online-serving policy (which generates the interaction data) and will not lead to any improvement. Therefore, we propose to decompose the problem into two sub-tasks, i.e., reconstruction and improvement, which effectively improve realizability and performance.
>
> ***
>
> ***Q3: What is the motivation/intuition of the adoption of a residual actor in the framework. Why and how the reward is decomposed into two parts. If we need the residual modeling, does it mean the first reconstruction step is inaccurate, which seems to contradict the assumption that sufficient data is accessible.***
>
> The motivation is that directly optimizing policy from scratch in the huge policy space is rather difficult, whereas improving upon a known policy (the online-serving policy) is relatively easy. Therefore, we propose to solve the problem by first imitating/reconstructing the online-serving policy and then refining the actions by adding a residual which is likely to achieve improvement. We do not decompose rewards. The first reconstruction step is to mimic the behaviors of the online-serving policy. After that, we improve the reconstructed policy by imposing the residuals. The residual module is optimized under the guidance of a state-value function through policy gradient, whose learning relies on the data collected by the online-serving policy.
>
> ***
>
> ***Q4: Are the classical reinforcement learning baselines performing any exploration in the experiments? If not, is the comparison valid?***
>
> The classical reinforcement learning baselines learn from the same interaction data collected by the online-serving policy, where the interaction data already contain diversity. The comparison is valid because random exploration (such as epsilon-greedy) in reinforcement learning is harmful to users’ experiences and is unrealistic to be introduced into recommendation. We cannot randomly perform a real recommendation and see how users will respond. This is dangerous and costly. Our evaluation method NCIS is a standard evaluation method in recommendation.

---

> > ### Author Response · Authors · 2022-11-25
> > **A gentle reminder**
> >
> > Dear Reviewer pqv6,
> >
> > We would like to know whether our responses have addressed your concerns. If you have any further questions, we are happy to discuss. Thank you.

---

### Decision · Program_Chairs · 2023-01-20

**Decision:**

Accept: poster

**Justification For Why Not Higher Score:**

This paper presents a solid piece of work but its novelty is slightly limited to be appealing to the general ICLR audience.

**Justification For Why Not Lower Score:**

I think it is basically unanimously agreed that the recommender systems research community would benefit from seeing this work. It has some flaws but not to the extent of being rejected.

**Metareview: Summary, Strengths And Weaknesses:**

Summary: This paper presents ResAct for sequential recommendation. The basic idea is to use offline reinforcement learning (RL) to optimize long-term engagement. As custom in many offline RL literature, ResAct starts by imitating the online serving policy and subsequently adding a residual (through a residual actor) to arrive at a policy that is close to but better than the online serving policy. ResAct also utilizes two information-theoretical regularizers to improve the expressiveness and conciseness of the extracted state features. Experimental results demonstrate that ResAct outperforms other RL approaches at optimizing long-term rewards.

Strengths: All the reviewers agree that this paper is addressing an important and challenging problem, which is to optimize for long-term engagement in sequential recommender systems. The proposed ResAct follows the standard offline RL approaches, that is to stay close to the logging policy and make a policy improvement. The paper is overall well-written (with minor issues below) and the experimental results are largely convincing (if with some minor issues, as pointed out by reviewer 2DQo).

Weaknesses: There are some concerns from the reviewers about the novelty of the proposed approach, which is a fair observation. However, I think this is overall a more applied work with promising empirical studies, which makes up for it. Another concern is around the two contributions of the paper (ResAct and the information-theoretical regularization) seemingly orthogonal to each other, which I agree, but even the reviewer who pointed it out doesn't think this is a major issue.

I also read the paper myself and have communicated my comments with the authors. To summarize, the authors take an unusual approach to use a continuous action space to represent items in a discrete set (which is also what makes adding a residual possible). This deserves much more elaboration (the authors did have a reasonable rationale for it). Furthermore, the paper claims "no need to explore" which is rather misleading (also pointed out by reviewer pqv6) and the authors agree to change the phrasing. Finally, this work draws close parallel to policy improvement work dating back to Kakade & Langford 2002 (with many followup work) which at least deserves some mentions.

**Note From Pc:**

if the above contains the word "oral" or "spotlight" please see: "oral" presentation means -> notable-top-5% and "spotlight" means -> notable-top-25%. As stated in our emails, we are disassociating presentation type from AC recommendations